# Prevalence and risks of tuberculosis multimorbidity in low-income and middle-income countries: a meta-review

Alexander Jarde ⦿ ,[1] Eugenia Romano,[2] Saima Afaq,[3,4] Asma Elsony,[5] Yan Lin,[6] Rumana Huque ⦿ ,[7] Helen Elsey ⦿ ,[1,8] Kamran Siddiqi ⦿ ,[1,8] B Stubbs,[2] Najma Siddiqi ⦿ [1,8]

[1]Department of Health Sciences, University of York, York, UK
[2]Department of Psychological Medicine, Institute of Psychiatry Psychology and Neuroscience, King's College London, London, UK
[3]Institute of Public Health and Social Sciences, Khyber Medical University, Peshawar, Pakistan
[4]Department of Epidemiology and Biostatistics, Imperial College London, London, UK
[5]Public Health, Epi-Lab, Khartoum, Sudan
[6]International Union Against Tuberculosis and Lung Disease, Beijing, China
[7]Economics, University of Dhaka, Dhaka, Bangladesh
[8]Hull York Medical School, Hull, UK

**Correspondence to**
Dr Alexander Jarde;
a.jarde@gmail.com

## ABSTRACT

**Objectives** Co-occurrence of tuberculosis (TB) with other chronic conditions (TB multimorbidity) increases complexity of management and adversely affects health outcomes. We aimed to map the prevalence of the co-occurrence of one or more chronic conditions in people with TB and associated health risks by systematically reviewing previously published systematic reviews.

**Design** Systematic review of systematic reviews (meta-review).

**Setting** Low-income and middle-income countries (LMICs).

**Papers** We searched in Medline, Embase, PsycINFO, Social Sciences Citation Index, Science Citation Index, Emerging Sources Citation Index and Conference Proceedings Citation Index, and the WHO Global Index Medicus from inception to 23 October 2020, contacted authors and reviewed reference lists. Pairs of independent reviewers screened titles, abstracts and full texts, extracted data and assessed the included reviews' quality (AMSTAR2). We included systematic reviews reporting data for people in LMICs with TB multimorbidity and synthesised them narratively. We excluded reviews focused on children or specific subgroups (eg, incarcerated people).

**Primary and secondary outcome measures** Prevalence or risk of TB multimorbidity (primary); any measure of burden of disease (secondary).

**Results** From the 7557 search results, 54 were included, representing >6 296 000 people with TB. We found that the most prevalent conditions in people with TB were depression (45.19%, 95% CI: 38.04% to 52.55%, 25 studies, 4903 participants, $I^2$=96.28%, high quality), HIV (31.81%, 95% CI: 27.83% to 36.07%, 68 studies, 62 696 participants, $I^2$=98%, high quality) and diabetes mellitus (17.7%, 95% CI: 15.1% to 20.0.5%, 48 studies, 48,036 participants, $I^2$=98.3%, critically low quality).

**Conclusions** We identified several chronic conditions that co-occur in a significant proportion of people with TB. Although limited by varying quality and gaps in the literature, this first meta-review of TB multimorbidity highlights the magnitude of additional ill health burden due to chronic conditions on people with TB.

**Prospero registration number** CRD42020209012.

## STRENGTHS AND LIMITATIONS OF THIS STUDY

⇒ We did an extensive search strategy, including databases of grey literature and protocols.

⇒ We summarised data synthesised at the country, regional (eg, Eastern Sub-Saharan Africa), continental and global (low-income and middle-income countries (LMICs)) level as long as the pooled estimate did not include data from high-income countries.

⇒ Whenever there was an overlap between two reviews in terms of countries covered, TB comorbidities and reported outcomes, we included the most complete one only if its quality, as assessed with the AMSTAR (a meaSurement tool to assess systematic reviews) 2 tool, was not lower than the other one.

⇒ Although we had initially planned to redo reported meta-analyses that included studies from high-income countries without these studies (to have pooled estimates from LMICs only), this was deemed unfeasible due to the high number of reviews where this would have been required.

## INTRODUCTION

About 30% adults in developed countries experience multimorbidity, that is, the co-occurrence of two or more chronic conditions (including non-communicable diseases (NCDs), chronic communicable diseases (CCDs) and mental disorders) in a single individual at one point in time. Multimorbidity is a growing global concern[1] and its prevalence is rising in low-income and middle-income countries (LMICs),[2] as CCDs such as tuberculosis (TB) and HIV remain major public health issues,[3] and NCDs are increasing due to major demographic shifts, urbanisation, changing environmental factors, economic empowerment and accompanying lifestyle changes.[4–8] This shift away from risks for CCD in children towards those for NCD in adults is also reflected in the steady increase in the burden of disability-adjusted life years (DALYs) attributed to NCDs over the past decades,[9] reaching 34% in low-income

countries, and up to 82% in middle-high-income countries in 2019.[10]

TB is one of the leading causes of mortality from a single infectious disease globally[9] and contributes 1.86% of the total worldwide DALYs and 2.54% of the total worldwide years of life lost (making it the 12th and 11th highest contributor, respectively).[4 10] TB frequently co-occurs with NCDs, including diabetes mellitus (DM, 2.79% of worldwide DALYs), depression (1.84% of worldwide DALYs) and cancer (neoplasms representing 9.93% of worldwide DALYs).[4 11] Depression[12] and DM[13] have been reported to be important risk factors for TB. Similarly, CCDs such as HIV (1.88% of worldwide DALYs) and TB adversely affect each other at the molecular, cellular, individual and population levels.[4 14]

We defined TB multimorbidity as the co-occurrence of TB and one or more chronic conditions (NCDs or CCDs).[15] This co-occurrence increases complexity of management and adversely affects health, economic and mortality outcomes, threatening the capacity for LMICs to achieve global public health targets. The cost and access to healthcare are of particular concern in LMICs, where the high costs relating to TB multimorbidity may further burden healthcare systems already under stress, and given the high out-of-pocket expenditure, it could lead to great financial burden for patients.

Numerous systematic reviews to date have considered individual chronic conditions in people with TB (eg, Huddart *et al*, Eshetie *et al*, Ruiz-Grosso *et al* and Gautam et al[16–19]). However, no review has synthesised the evidence on a range of chronic conditions, their prevalence in people with TB and the burden associated with such co-occurrence of conditions. Understanding the overarching literature on TB multimorbidity is essential to enable better services to be developed to identify, prevent and manage this common situation, which presents a significant health and financial burden to people with TB and to health services. Furthermore, differences in TB multimorbidity by gender, socio-economic group and country, which could shed further light on the problem, remain unclear.

The primary aim of this comprehensive meta-review of systematic reviews was to summarise and map the prevalence and risk of chronic conditions (CCD or NCD, alone or in combination) in people with TB in LMICs compared with people without TB, and to summarise the associated health outcomes (eg, TB treatment success and measures of disease burden) in people with TB multimorbidity, compared with people with TB only.

## Methods
We have followed the Preferred Reporting Items for Systematic Reviews and Meta-Analyses guidelines[20] in reporting this meta-review and its protocol was registered in the international prospective register of systematic reviews (PROSPERO, CRD42020209012).

## Search strategy
We ran our search strategy in Medline (Ovid), Embase (Ovid), PsycINFO (Ovid), Social Sciences Citation Index (Web of Science), Science Citation Index (Web of Science), Emerging Sources Citation Index and Conference Proceedings Citation Index (Web of Science) and the WHO Global Index Medicus from inception to 23 October 2020. To identify unpublished studies, we also searched PROSPERO and the Open Grey database, and contacted authors of conference abstracts. Reference lists of included reviews were hand searched. We did not set any restrictions on the origin of the paper, date of publication or language.

We used free text and controlled vocabulary (eg, Medical Subject Headings [MeSH] terms for Medline) for terms related to communicable, non-communicable and mental diseases and combined them with terms for TB using Boolean operators: (CCD or NCD or mental disease) AND Tuberculosis. Online supplemental appendix 1 lists the search terms for Medline and the full search strategy can be found in online supplemental appendix 2.

## Selection criteria
We included systematic reviews reporting data for people in LMICs, with any type of TB and one or more additional chronic conditions. This included, but was not limited to, heart disease, DM, arthritis, chronic obstructive pulmonary disease, HIV, Hepatitis B (HBV) and Hepatitis C (HCV), depression and anxiety disorders (as defined by review authors). As there is no clear and widely used definition of what constitutes a chronic condition,[21] whenever there were doubts, four of the authors with clinical/research expertise (KS, NS, HE and BS) decided by consensus if a disorder was to be considered as a chronic condition. Conditions considered side effects of TB medications, such as nausea or diarrhoea, were not considered chronic conditions for this review.

After registering the protocol, the following additional changes were made. First, we decided to limit our population of interest to the general TB population, excluding studies that stated focusing on children. Second, we decided to exclude studies that focused on specific subgroups (eg, incarcerated people, healthcare workers, etc), focussing on populations for which results are more readily generalisable. Studies in patients with a specific type of TB (eg, extra-pulmonary TB) were, however, considered eligible.

Included systematic reviews had to report either pooled or individual study data for at least one of our primary or secondary outcomes. Narrative, non-systematic reviews and systematic reviews focused only on high-income countries (HICs) were excluded.

## Primary outcomes
The coprimary outcomes included prevalence (or incidence) of each chronic condition (or combination of more than one condition) in people with TB, and odds ratios (or other comparative statistic) of having a chronic condition (or combination of conditions) in people with TB compared with those without TB.

## Secondary outcomes

Secondary outcomes included any measure of disease burden in people with TB multimorbidity, such as mortality, loss to follow-up (treatment interrupted for two consecutive months or more), treatment failure (sputum smear or culture remained positive at month 5 or later during treatment), treatment completion (without evidence of failure, but with no record of being cured), cured (smear-negative or culture-negative patients in the last month of treatment and on at least one previous occasion), successful treatment (patients who were cured or who completed treatment) or unsuccessful treatment (patients who were lost to follow-up, had treatment failure or died).[22 23] Other secondary outcomes of interest included years of life lived with disability, years of life lost, DALYs, outcomes related to the additional chronic conditions and any other reported measure of disease burden.

## Study selection

Multiple authors (ER, SA, AJ and NS) contributed to the screening and data extraction procedures, with titles and abstracts of all deduplicated search results screened independently by at least two reviewers. The full text of potentially eligible papers was reviewed against our inclusion and exclusion criteria independently by two reviewers. Disagreements were resolved by discussion, with a third reviewer available as an arbitrator if necessary. We used the online software Rayyan (https://rayyan.ai/) to manage the study selection process.

## Data extraction and quality assessment

Two reviewers used a piloted form (Google Form) developed for the review to independently extract data regarding review characteristics, characteristics of included primary studies and outcome data. If clarifications were needed, we contacted the corresponding authors.

The quality of included systematic reviews was assessed by two reviewers (ER and SA, with discrepancies resolved by agreement or a third independent assessor, AJ) using the AMSTAR (A MeaSurement Tool to Assess systematic Reviews) 2 tool, which classifies the overall confidence in the results of each review as critically low, low, moderate or high.[24]

## Data synthesis

The following steps were followed to synthesise the evidence. First, all included systematic reviews were described in a summary table. Second, the results (primary and secondary outcomes) for each combination of conditions were summarised, including the pooled estimates, the number of studies, pooled sample size, a measure of heterogeneity, range of pooled effect sizes and quality assessment. Third, the results were stratified by age, gender, socioeconomic group, type of TB and region, where possible. We had initially planned to extract and pool individual study data for LMICs when such studies had been pooled together with data from HICs, or when individual study data were reported but not pooled in a meta-analysis. However, such an approach was deemed unfeasible due to the high number of reviews where this would have been required. In these cases, we reported the study characteristics and the range of study effect sizes from LMICs.

## Patient and public involvement

We asked patients' representatives for feedback on the study protocol and they will be involved in the dissemination of our results. Patients or the public were not involved in the conduct or reporting of our research.

The funder of the study had no role in study design, data collection, data analysis, data interpretation, or writing of the report.

## RESULTS

Our search strategy identified 7557 results, of which 2200 were duplicates and were removed. Of the 221 results remaining after screening titles and abstracts, 130 were excluded for not meeting eligibility criteria. Online supplemental appendix table 2 specifies the reasons for exclusion. The full text corresponding to 34 protocols or conference abstracts could not be obtained. We contacted the authors of these references (with a follow-up email 2 weeks later), seven of them replied confirming that no full article had been published. Three journal articles, related to coronary heart disease, head and neck TB, and HBV, could not be assessed in full text despite our efforts (no institutional access and no response from authors[25–27]). The full text of one additional study[25] could not be obtained, but the pooled relative risk of coronary heart disease was reported in the abstract and was therefore included. Ultimately, 54 studies were included in our review (figure 1).

## Study and participant characteristics

Overall, there were over 6 296 000 people with TB across the 54 included systematic reviews, covering 85 LMICs (Appendix 3). Of these, 23 reported a pooled estimate of interest to our review (S1–S23), while the remainder reported outcomes of interest for individual studies, but either did not pool them in a meta-analysis or pooled them with data from HICs. Among the 23 reviews reporting pooled outcomes, even when they assessed the same combination of TB and chronic condition(s), there was limited overlap between them with regards to geographical region and/or reported outcomes (online supplemental appendix table 3). Online supplemental appendix table 4 details outcome information reported by each review.

Most of the included systematic reviews reported data on TB without specifying a particular type of TB (S1–S4, S7–S10, S14, S16, S17, S20–S46); nine focused on drug-resistant TB (DR-TB) (S6), multidrug-resistant TB (MDR-TB) (S5, S12, S19, S47–S50) or extensively drug-resistant TB (XDR-TB) (S12, S47); three focused

 

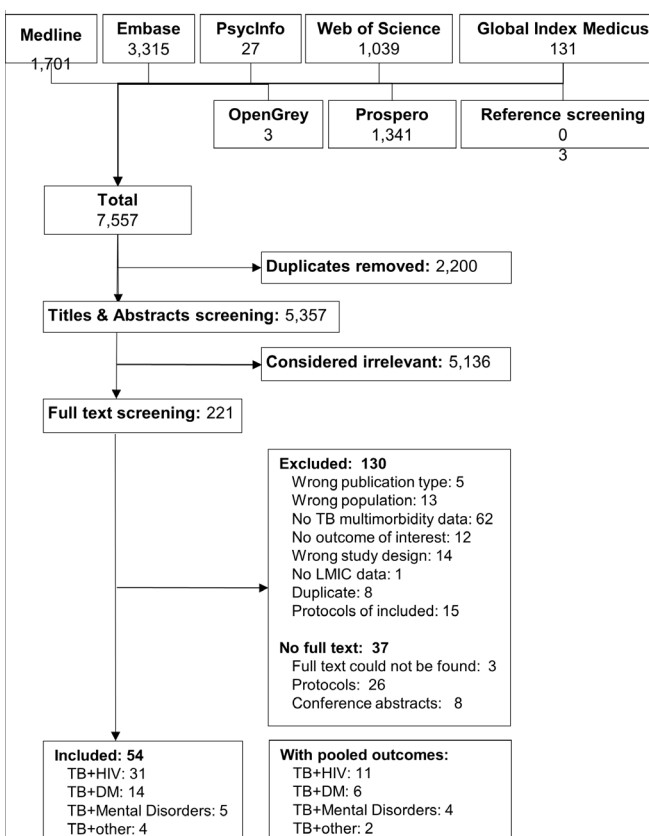

**Figure 1** Flow diagram of the search results and screening process. DM, diabetes mellitus; LMICs, low-income and middle-income countries; TB, tuberculosis.

on pulmonary TB (PTB) (S15, S18, S51), three on TB meningitis (S13, S52, S53) and one on TB lymphadenitis (S11). The chronic conditions most often considered were HIV (31 reviews),(S1, S2, S5–S13, S24–S36, S47–S50, S52–S54) DM (14 reviews) (S4, S14–S18, S37–S43, S51) and mental illness (five reviews) (S3, S19–S21, S44). None of the systematic reviews reported results on the prevalence and/or associated risks of more than one additional chronic condition in people with TB. Online supplemental appendix table 5 lists what conditions were considered or not a chronic condition for this review.

Most of the identified systematic reviews were assessed as low or critically low quality according to AMSTAR2 (n=42). Only seven reviews were assessed as moderate (n=2)(S7, S22) or high (n=5) (S6, S10, S11, S20, S42) quality, six of which reported a pooled estimate of interest. The critical domains that failed most often were regarding risk of bias assessment (37 studies) and protocol registration (29 studies). Online supplemental appendix table 6 details the AMSTAR2 assessment for each study.

## Summary of results
### TB and HIV
Of the 31 reviews reporting data on TB and HIV (>3 017 000 participants from 72 countries) (S1, S2, S5–S13, S24–S36, S47–S50, S52–S54), 11 focused on specific types of TB (S5, S6, S11–S13, S47–S50, S52, S53) and 11 reported

at least one pooled outcome of interest (online supplemental appendix table 3) (S1, S2, S5–S13).

One review (S9) reported the pooled prevalence for Latin America (25%, 95% CI: 19.3% to 30.8%, 7 studies, critically low quality) and Africa (31.2%, 95% CI: 19.3% to 43.2%, 17 studies, critically low quality). Prevalence estimates for subcontinental regions were also reported in other reviews, ranging from 25% in Western Sub-Saharan Africa (SSA, high quality) to 44% in Southern SSA (high quality), as well as for China, Ethiopia and Iran (table 1).

One review (S5) reported a reduced odds of treatment success (OR: 0.87, 95% CI: 0.79 to 0.96, 6 studies, critically low quality) in people with TB and HIV compared with people with only TB, in SSA.

Table 1 also summarises the results of systematic reviews reporting data for specific types of TB (DR-TB, MDR-TB, PTB, TB meningitis and TB lymphadenitis).

### TB and DM
Of the 14 reviews reporting data on TB and DM (>2 878 000 participants from 48 countries) (S4, S13–S17, S36–S42, S50), three focused on specific types of TB (S14, S17, S50) and six reported at least one pooled outcome of interest (online supplemental appendix table 3) (S4, S14–S18).

One review (S17) reported the pooled prevalence separately for low-income countries (7.9%, 95% CI: 4.9% to 11.5%, 15 studies, 9434 participants, critically low quality), lower-middle income countries (17.7%, 95% CI: 15.1% to 20.5%, 48 studies, 48 036 participants, critically low quality) and upper-middle income countries (14.4%, 95% CI: 12.8% to 16.0%, 75 studies, 1 994 027 participants, critically low quality). The same review also reported the prevalence of DM in people with TB in Africa (8.0%, 95% CI: 5.9% to 10.4%, 119 studies, 474 944 participants, critically low quality, table 2. Pooled prevalences in other continents were also reported, but were excluded from our review, as they included data from HICs. Other reviews reported prevalence estimates for subcontinental regions, ranging from 9% in SSA (low quality) (S14) to 21% in South Asia (low quality) (S4) as well as for multiple individual countries (figure 2, table 2).

One review (S16) reported an increased odds of mortality (OR: 1.80, 95% CI: 1.35 to 2.40, 34 studies, low quality) and treatment failure or death (OR: 1.90, 95% CI: 1.43 to 2.53, 22 studies, low quality) in people with TB and DM compared with people with only TB, in LMICs overall.

Table 2 also summarises the results of systematic reviews focused on (or reporting data for) specific types of TB (MDR-TB and PTB).

### TB and mental disorders
*TB and mental disorders (pooled as a composite outcome)*
We found one systematic review considering a composite outcome for mental disorders (S21), as well as several other reviews looking at individual mental disorders such

**Table 1** Prevalence of HIV and effect on outcomes in people with TB

| Region | Outcome (quality*) | Effect size (95% CI) | Range | Number of studies (number of participants) | I² | Year |
|---|---|---|---|---|---|---|
| Latin America | Prevalence (critically low) (S9) | 25% (19.3% to 30.8%) | NR | 7 (NR) | 95.2% | 2013 |
| Africa | Prevalence (critically low) (S9) | 31.2% (19.3% to 43.2%) | NR | 17 (NR) | 99.6% | 2013 |
| SSA | Prevalence (high) (S10) | 31.81% (27.83% to 36.07%) | 6.03%–72.25% | 68 (62 696) | 98% | 2019 |
| | Successful treatment (critically low) (S5) | OR: 0.87 (0.79 to 0.96) | 0.75–1.26 | 6 (NR) | NR | 2019 |
| Western SSA | Prevalence (high) (S10) | 25.48% (19.70% to 32.27%) | 10.26%–72.13% | 21 (16 145) | 98% | 2019 |
| Eastern SSA | Prevalence (high) (S10) | 31.14% (25.39% to 37.54%) | 6.03%–60.51% | 32 (33 637) | 98% | 2019 |
| Southern SSA | Prevalence (high) (S10) | 43.67% (35.05% to 52.69%) | 23.84%–72.25% | 12 (11 148) | 99% | 2019 |
| Central SSA | Prevalence (high) (S10) | 41.33% (30.39% to 53.19%) | 31.29%–51.56% | 3 (2039) | 96% | 2019 |
| China | Prevalence (critically low) (S8) | 0.9% (0.6% to 1.4%) | 0.1%–4.5% | 18 (NR) | 92.21% | 2010 |
| | Prevalence in women (critically low) (S8) | 0.6% (0.3% to 1.1%) | NR | 9 (NR) | 71.8% | 2010 |
| | Prevalence in men (critically low) (S8) | 1.1% (0.6% to 2.0%) | NR | 9 (NR) | 94.7% | 2010 |
| Ethiopia | Prevalence (moderate) (S7) | 23.40% (19.56% to 27.24%) | 9.50%–52.10% | 13 (19 212) | 97.6% | 2019 |
| | Successful treatment (critically low) (S2) | 67% (56% to 79%) | NR | NR (NR) | NR | 2018 |
| | Unsuccessful treatment (critically low) (S2) | 33% (21% to 44%) | NR | NR (NR) | NR | 2018 |
| | Unsuccessful treatment (critically low) (S2) | OR: 1.98 (1.56 to 2.52) | 0.82–14.31 | 20 (NR) | 81.0% | 2018 |
| India | Case-fatality rate during treatment (critically low) (S1) | 10.91% (7.68% to 15.50%)† | NR | 35 (NR) | Tau²=0.90‡ | 2020 |
| | Case-fatality rate after treatment (critically low) (S1) | 4.15% (1.06% to 16.24%) | NR | 5 (NR) | Tau²=1.90‡ | 2020 |
| Iran | Prevalence (critically low) (S28) | 14% (12% to 15%) | 0%–54% | 48 (21 388) | 97.93% | 2019 |
| **Drug-resistant TB + HIV** | | | | | | |
| SSA | Unsuccessful treatment (high) (S6) | RR: 1.18 (1.07 to 1.30)‡·§ | 0.71–2.37 | 19 (8301) | 48% | 2020 |
| | Treatment failure (high) (S6) | RR: 0.66 (0.38 to 1.13)§ | 0.15–2.40 | 10 (5474) | 73% | 2020 |
| | Loss to follow-up (high) (S6) | RR: 0.82 (0.74 to 0.92) § | 0.49–2.61 | 14 (7051) | 0% | 2020 |
| | Mortality (high) (S6) | RR: 1.50 (1.30 to 1.74)§ | 0.73–2.18 | 16 (7365) | 39% | 2020 |
| Western SSA | Unsuccessful treatment (high) (S6) | RR: 1.42 (0.95 to 2.13) | 1.31–2.37 | 2 (790) | 12% | 2020 |
| | Mortality (high) (S6) | RR: 1.42 (0.96 to 2.09) | NA | 1 (588) | NA | 2020 |
| Eastern SSA | Unsuccessful treatment (high) (S6) | RR: 1.47 (1.23 to 1.75) | 1.14–1.77 | 6 (1970) | 0% | 2020 |
| | Mortality (high) (S6) | RR: 1.52 (1.19 to 1.93) | 1.20–2.18 | 5 (1442) | 0% | 2020 |
| Southern SSA | Unsuccessful treatment (high) (S6) | RR: 1.09 (0.98 to 1.20)§ | 0.71–1.41 | 11 (5541) | 43% | 2020 |
| | Mortality (high) (S6) | RR: 1.49 (1.21 to 1.83)§ | 0.73–1.47 | 10 (5335) | 60% | 2020 |
| **Multidrug-resistant TB + HIV** | | | | | | |
| LMICs | Unsuccessful treatment (low) (S12) | RR: 1.34 (1.04 to 1.72) | 0.55–3.33 | 13 (5816) | 88% | 2018 |
| Low income countries | Unsuccessful treatment (low) (S12) | RR: 2.23 (1.60 to 3.11) | 0.67–3.33 | 7 (NR) | 41% | 2018 |
| | Treatment failure (low) (S12) | RR 0.75 (0.44 to 1.29) | 0.32–2.40 | 7 (5930) | 55% | 2018 |

Continued

**Table 1** Continued

| Region | Outcome (quality*) | Effect size (95% CI) | Range | Number of studies (number of participants) | I² | Year |
|---|---|---|---|---|---|---|
| SSA | Successful treatment (critically low) (S5) | OR 0.87 (0.79 to 0.96) | 0.75–1.26 | 6 (NR) | NR | 2019 |
|  | Mortality (critically low) (S5) | 18% (14% to 23%) | 9%–31% | 9 (NR) | 91.1% | 2019 |
|  | Cured (critically low) (S5) | 34% (22% to 45%) | 3%–60% | 9 (NR) | 98.9% | 2019 |
| **Pulmonary TB + HIV** |  |  |  |  |  |  |
| Ethiopia | Prevalence (moderate) (S7) | 22.08% (14.36% to 29.81%) | 4.97%–28.60% | 3 (1079) | 89.9% | 2019 |
| **TB meningitis + HIV** |  |  |  |  |  |  |
| LMICs | Prevalence (critically low) (S13) | 10.6% (4.2% to 24.6%) | NR | NR (NR) | NR | 2019 |
|  | Mortality (critically low) (S13) | 53.4% (42.4% to 64.1%) | NR | 7 (547) | 2.1% | 2019 |
| **TB lymphadenitis + HIV** |  |  |  |  |  |  |
| Africa | Prevalence (high) (S11) | 52% (33% to 71%) | 6%–91% | 14 (NR) | 99.2% | 2019 |
| Ethiopia | Prevalence (high) (S11) | 21% (12% to 30%) | 6%–67% | 6 (NR) | 92.9% | 2019 |

*Quality rating represents the overall confidence in the results of the review, as assessed with the AMSTAR2 tool.
†Including only seven high-quality studies: 12.17% (95% CI: 5.68% to 26.11%).
‡In the original review they considered this value low if it was <4.
§Includes one study focused on children.
¶Excluding one study focused on children: RR: 1.12 (0.96 to 1.30, I2=69%).
LMICs, low-income and middle-income countries; NR, not reported; SSA, Sub-Saharan Africa; TB, tuberculosis.

as depression, anxiety and psychosis. The review (S21) that reported the effect of mental disorders (defined as a composite variable including depression, psychological distress, Post-traumatic stress disorder [PTSD] or mental disorder) on unsuccessful treatment (a composite measure combining some or all of treatment failure, loss to follow-up and death), loss to follow-up and non-adherence, found no evidence of a significant increase in the odds of these outcomes in people with TB and mental disorders, compared with people with only TB (table 3).

### TB and depression
Of the four reviews reporting data on TB and depression (>21 770 participants from 33 countries) (S3, S19, S20, S44), three (S3, S20, S44) reported at least one pooled outcome of interest (online supplemental appendix table 3). One systematic review (S20) of 25 studies reported the prevalence of depression in people with TB in LMICs as 45.19% (95% CI: 38.04% to 52.55%, 25 studies, 4903 participants, high quality). None of the included reviews reported this outcome at a continental, regional or country level (table 3). One systematic review (S3) reported an increased odds of mortality (OR: 2.85, 95% CI: 1.52 to 5.36, 2 studies, 1303 participants, critically low quality) and other adverse outcomes in people with TB and depression compared with people with only TB (table 3). Table 3 summarises the results of systematic reviews focused on MDR-TB. According to these results, the prevalence of depression in people with MDR-TB is 52% (95% CI: 38% to 66%, 5 studies, high quality) (S20).

### TB and anxiety
Of the two (S19, S44) reviews reporting data on TB and anxiety (>7500 participants from 31 countries), only one (S19) focused on MDR-TB, reported any pooled outcome of interest: the prevalence of anxiety overall (24%, 95% CI: 2% to 57%, 3 studies, 209 participants, critically low quality) and in the regions of Southeast Asia and the Americas (table 3).

### TB and psychosis
One systematic review (7518 participants from 17 countries) focused on MDR-TB, reported the prevalence of psychosis in Africa (12%, 95% CI: 8% to 17%, 5 studies, critically low quality) and in several subcontinental regions (table 3) (S19).

### TB and HCV
One systematic review estimated the prevalence of HCV in people with TB in Africa to be 11% (95% CI: 1% to 23%, 3 studies, 327 participants, I²=93.9%, moderate quality) (S22).

### Risk of cancer in people with TB
One systematic review (S23) reported the risk of different types of cancer in people with TB in upper-middle income countries, including lung cancer (RR: 1.53, 95% CI: 1.25 to 1.87, 9 studies, low quality), non-Hodgkin's lymphoma (RR: 1.70, 95% CI: 1.13 to 2.56, 1 study, low quality) and

**Table 2** Prevalence of DM and effect on outcomes in people with TB

| Region | Outcome (quality*) | Effect size(95% CI) | Range | Number of studies (number of participants) | I² | Year |
|---|---|---|---|---|---|---|
| LMICs | Mortality (low) (S16) | OR: 1.80 (1.35 to 2.40) | 0.45–29.22 | 34 (NR) | 91.1% | 2019 |
| | Treatment failure or death (low) (S16) | OR: 1.90 (1.43 to 2.53) | 0.73–11.75 | 22 (NR) | 87.3% | 2019 |
| World (low-income countries) | Prevalence (critically low) (S17) | 7.9% (4.9% to 11.5%) | NR | 15 (9434) | 96.8% | 2019 |
| World (lower-middle income countries) | Prevalence (critically low) (S17) | 17.7% (15.1% to 20.5%) | NR | 48 (48 036) | 98.3% | 2019 |
| World (upper-middle income countries) | Prevalence (critically low) (S17) | 14.4% (12.8% to 16.0%) | NR | 75 (1 994 027) | 99.9% | 2019 |
| Africa | Prevalence (critically low) (S17) | 8.0% (5.9% to 10.4%) | 1.9%–32.4% | 119 (474 944) | 99.8% | 2019 |
| Southeast Asia | Prevalence (critically low) (S 17) | 19.0% (16.2% to 21.9%) | 5.1%–54.1% | 30 (30 382) | 97.0% | 2019 |
| South Asia | Prevalence (low) (S4) | 21% (18% to 23%) | 4%–66% | 65 (NR) | 98.28% | 2021 |
| SSA | Prevalence (low) (S14) | 9% (6% to 12%) | 2%–38% | 16 (13 286) | 97.48% | 2019 |
| Bangladesh | Prevalence (critically low) (S17) | 10.6% (7.2% to 14.5%) | 8.3%–12.8% | 3 (3010) | 85.9% | 2019 |
| Benin | Prevalence (critically low) (S17) | 1.9% (0.2% to 4.7%) | NA | 1 (159) | NA | 2019 |
| Brazil | Prevalence (critically low) (S17) | 7.2% (6.3% to 8.1%) | 3.3%–33.1% | 12 (1 726 436) | 99.7% | 2019 |
| China | Prevalence (critically low) (S17) | 14.5% (10.5% to 19.0%) | 2.7%–30.1% | 14 (19 529) | 98.4% | 2019 |
| Egypt | Prevalence (critically low) (S17) | 22.8% (15.2% to 31.4%) | 15.8%–27.7% | 3 (578) | 81.4% | 2019 |
| Ethiopia | Prevalence (critically low) (S17) | 18.8% (1.9% to 47.1%) | 8.3%–32.4% | 2 (1749) | 99.2% | 2019 |
| Fiji | Prevalence (critically low)(S17) | 10.1% (4.4% to 17.7%) | 5.2–13.7 | 3 (1139) | 91.8% | 2019 |
| Georgia | Prevalence (critically low)(S17) | 12.4% (7.4% to 18.5%) | NA | 1 (137) | NA | 2019 |
| Guinea-Bissau | Prevalence (critically low) (S17) | 2.7% (0.3% to 6.8%) | NA | 1 (110) | NA | 2019 |
| Guyana | Prevalence (critically low) (S17) | 14.0% (7.8% to 21.6%) | NA | 1 (100) | NA | 2019 |
| India | Prevalence (low) (S4) | 22.0% (19.0% to 25.0%) | NR | 47 (NR) | 97.92% | 2021 |
| | Mortality (low) (S4) | OR: 1.74 (1.21 to 2.51) | NR | 5 (NR) | 19.43% | 2021 |
| | Treatment failure (low) (S4) | OR: 1.65 (1.12 to 2.44) | NR | 5 (NR) | 49.63% | 2021 |
| | Recurrence (low) (S4) | OR: 0.53 (0.32 to 0.87) | NA | 1 (NR) | NA | 2021 |
| | Cured (low) (S4) | OR: 0.32 (0.10 to 1.05) | NA | 1 (NR) | NA | 2021 |
| Indonesia | Prevalence (critically low) (S17) | 14.8% (12.2% to 17.7%) | NA | 1 (634) | NA | 2019 |
| Iran | Prevalence (critically low) (S17) | 17.8% (12.5% to 23.8%) | 5.5%–40.0% | 11 (3134) | 93.3% | 2019 |
| Kazakhstan | Prevalence (critically low) (S17) | 7.1% (5.1% to 9.4%) | NA | 1 (562) | NA | 2019 |
| Kiribati | Prevalence (critically low) (S17) | 36.7% (31.1% to 42.5%) | NA | 1 (275) | NA | 2019 |
| Libya | Prevalence (critically low) (S17) | 6.1% (3.5% to 9.4%) | NA | 1 (262) | NA | 2019 |
| Malaysia | Prevalence (critically low) (S17) | 26.9% (17.8% to 37.0%) | 15.4%–39.0% | 5 (23 438) | 98.1% | 2019 |
| Marshall Islands | Prevalence (critically low) (S17) | 45.2% (32.9% to 57.7%) | NA | 1 (62) | NA | 2019 |
| Mexico | Prevalence (critically low) (S17) | 30.8% (26.4% to 35.3%) | 19.3%–54.4% | 10 (192 420) | 97.9% | 2019 |
| Nepal | Prevalence (low) (S4) | 12.0% (4.0% to 20.0%) | NR | 4 (NR) | 97.6% | 2021 |
| Nigeria | Prevalence (critically low) (S17) | 7.8% (4.4% to 12.0%) | 4.8%–12.0% | 4 (9821) | 97.8% | 2019 |
| Pakistan | Prevalence (low) (S4) | 19.0% (11.0% to 27.0%) | NR | 10 (NR) | 99.18% | 2021 |

Continued

**Table 2** Continued

| Region | Outcome (quality*) | Effect size(95% CI) | Range | Number of studies (number of participants) | I² | Year |
|---|---|---|---|---|---|---|
| Peru | Prevalence (critically low) (S17) | 4.8% (1.7% to 9.5% | 2.5%–11.1% | 4 (3983) | 96.8% | 2019 |
| Romania | Prevalence (critically low) (S17) | 18.4% (13.6% to 23.7%) | NA | 1 (228) | NA | 2019 |
| Senegal | Prevalence (critically low) (S17) | 4.9% (2.2% to 8.5%) | 3.8%–7.0% | 2 (2848) | 75.1% | 2019 |
| South Africa | Prevalence (critically low) (S17) | 9.4% (7.6% to 11.3%) | NA | 1 (947) | NA | 2019 |
| Sri Lanka | Prevalence (low) (S4) | 24.0% (21.0% to 27.0%) | NR | 2 (NR) | NR | 2021 |
| Tanzania | Prevalence (critically low) (S17) | 8.5% (4.8% to 13.0%) | 2.6%–16.7% | 7 (4178) | 95.1% | 2019 |
| Thailand | Prevalence (critically low) (S17) | 7.5% (6.2% to 8.8%) | 6.0%–16.3% | 5 (17 862) | 81.6% | 2019 |
| Tunisia | Prevalence (critically low) (S17) | 7.6% (5.9% to 9.6%) | NA | 1 (788) | NA | 2019 |
| Turkey | Prevalence (critically low) (S17) | 7.8% (6.8% to 8.8%) | 7.9%–8.6% | 3 (2773) | 0% | 2019 |
| Uganda | Prevalence (critically low) (S17) | 7.3% (4.7% to 10.3%) | 5.4%–8.5% | 2 (390) | 9.9% | 2019 |
| Yemen | Prevalence (critically low) (S17) | 9.5% (6.0% to 13.8%) | NA | 1 (220) | NA | 2019 |
| Multidrug-resistant TB + DM | | | | | | |
| LMICs | Unsuccessful treatment (low) (S12) | RR: 0.90 (0.65 to 1.23) | 0.23–0.98 | 3 (687) | 19% | 2018 |
| Pulmonary TB + DM | | | | | | |
| China | Prevalence (critically low) (S15) | 7.20% (6.01% to 8.39%) | 2.08%–16.16% | 22 (56 805) | NR | 2013 |
| | Retreatment (critically low) (S18) | OR: 2.05 (1.30 to 3.22) | NR | 3 (499) | 0% | 2016 |
| | Retreatment (critically low) (S18) | aOR: 3.38 (1.56 to 7.29) | NR | 2 (NR) | 75% | 2016 |

*Quality rating represents the overall confidence in the results of the review, as assessed with the AMSTAR2 tool.
aOR, adjusted odds ratio; DM, diabetes mellitus; LMICs, low-income and middle-income countries; NR, not reported; SSA, Sub-Saharan Africa; TB, tuberculosis.

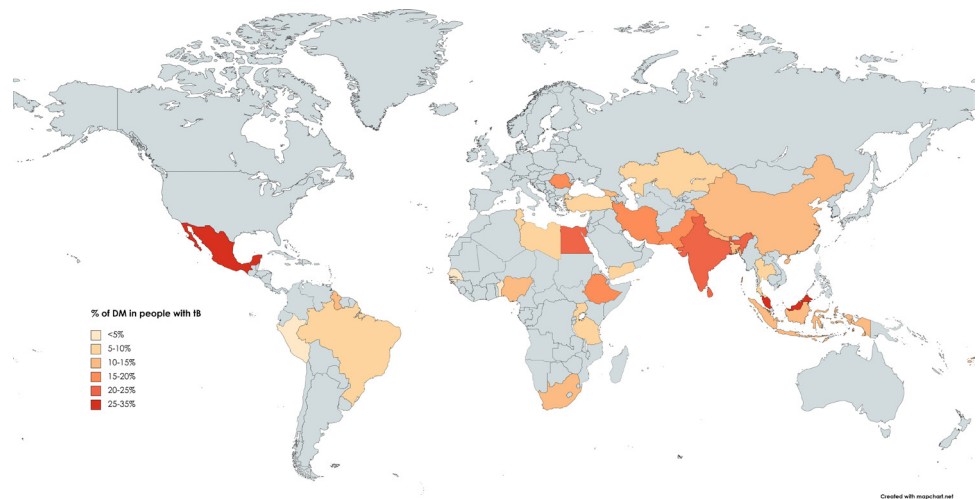

**Figure 2** Prevalence of DM in people with TB in each country. DM, diabetes mellitus; TB, tuberculosis.

leukaemia (RR: 1.61, 95% CI: 1.13 to 2.29, 1 study, low quality) (table 4).

### Risk of coronary heart disease in people with TB

One systematic review reported (in their abstract) an increased risk of coronary heart disease in people with TB in LMICs (RR: 1.76, 95% CI: 1.05 to 2.95) (table 4) (S55).

### Subgroup analyses

Regarding our planned subgroup analyses, we could only find data stratified by gender reported for the prevalence of HIV in people with TB in China (women: 0.6%, 95% CI: 0.3% to 1.1%, 9 studies, critically low quality; men: 1.1%, 95% CI: 0.6% to 2.0%; 9 studies, critically low quality) (table 1) (S8) and the prevalence of depression in people with TB (women: 51.54%, 95% CI: 40.34% to 62.60%, 17 studies, high quality; men: 45.25%, 95% CI: 35.19% to 55.71%, 17 studies, high quality) (table 3) (S20) We did not find any pooled results stratified by age.

### DISCUSSION

This was the first meta-review to identify and map out the co-occurrence of CCDs and NCDs in people with TB in LMICs. Although the geographical regions covered by the included reviews varied, we found that the most prevalent chronic conditions were depression, HIV and DM. We also found some evidence that people with TB and these chronic conditions had significantly increased odds of adverse outcomes such as death and treatment failure. No systematic review pooled the prevalence of two or more additional chronic conditions in people with TB and differences between people with TB and a single chronic condition versus multiple additional chronic conditions could not be explored.

While HIV, DM and depression are well-known comorbidities of TB, our review highlights that their prevalence can vary, in some cases substantially, between different countries or regions. Such regional differences should be taken into account when designing interventions,

illustrating how a one-size-fits-all approach is unlikely to succeed.

Our findings offer an overview of TB multimorbidity to see comorbid conditions in relation to each other. For instance, despite the known synergistic relationship between TB and HIV,[14] our review suggests that the negative impact of HIV on TB treatment outcomes is less severe than the impact of depression, which not only had higher odds of adverse outcomes, but also was more prevalent among people with TB. This apparent smaller impact of HIV than depression in people with TB could partially be explained by the disparity—in attention and resources—between HIV and depression, and illustrates how an integrated approach, such as the one received by at least some patients with TB and HIV, could reduce the negative impact of other chronic conditions, such as depression, in patients with TB. This also illustrates how the results of our review could be used when planning for new services. Moreover, it highlights the importance of screening for mental health in areas where mental health services need improvement.[28 29]

Our meta-review highlights the many gaps in the literature on TB multimorbidity in LMICs. For example, while the meta-analysis of the prevalence of TB and depression included 25 studies,[30] the meta-analysis for treatment outcomes in this group included only two studies,[18] reflecting the lack of evidence for the impact of TB multimorbidity on TB treatment outcomes. Data stratified by gender or age were also minimal, which is particularly important when women might have different healthcare seeking behaviours and limited voice in decision-making. In addition to the gaps in the literature with regards to primary studies, our meta-review also highlights the lack of systematic reviews focused on people with TB and more than one additional chronic condition, which is an increasingly likely scenario as the prevalence of NCDs in LMICs grows.[2] In this regard, several cohort studies have assessed the impact of multimorbidity on TB treatment outcomes, such as in China or Brazil[31 32] finding

**Table 3** Prevalence and effect on outcomes in people with TB and mental disorders: as a composite outcome and separately for depression, anxiety and psychosis

| Region | Outcome (quality)* | Effect size (95% CI) | Range | Number of studies (number of participants) | I² | Year |
|---|---|---|---|---|---|---|
| **TB + mental disorders (composite measure)†** | | | | | | |
| LMICs | Unsuccessful treatment‡ (critically low) (S21) | OR: 2.13 (0.85 to 5.37) | 0.80–4.25 | 4 (1196) | 82% | 2020 |
| | Loss to follow-up (critically low) (S21) | OR: 1.90 (0.33 to 10.91) | 0.88–5.33 | 2 (1139) | 78% | 2020 |
| | Non-adherence to treatment (critically low)(S21) | OR: 1.60 (0.84 to 3.02) | 0.94–3.67 | 4 (10 851) | 86% | 2020 |
| **TB + depression** | | | | | | |
| LMICs | Prevalence (high) (S20) | 45.19% (38.04% to 52.55%) | 15.56%–80.00% | 25 (4903) | 96.28% | 2020 |
| | Prevalence in women (High) (S20) | 51.54% (40.34% to 62.60%) | NR | 17 (NR) | 92.55% | 2020 |
| | Prevalence in men (high) (S20) | 45.25% (35.19% to 55.71%) | NR | 17 (NR) | 95.09% | 2020 |
| | Mortality or loss to follow-up (critically low) (S3) | OR: 4.26 (2.33 to 7.79) | 3.65–4.88 | 2 (1303) | 0% | 2020 |
| | Mortality (critically low) (S3) | OR: 2.85 (1.52 to 5.36) | 1.76–2.99 | 2 (973) | 0% | 2020 |
| | Loss to follow-up (critically low) (S3) | OR: 8.70 (4.95 to 9.09) | 4.95–9.09 | 2 (973) | 0% | 2020 |
| | Non-adherence to TB treatment (critically low) (S3) | OR: 1.38 (0.70 to 2.72) | 0.92–3.67 | 3 (9349) | 94.36% | 2020 |
| **Multidrug-resistant TB + depression** | | | | | | |
| LMICs | Prevalence (high) (S20) | 52.34% (38.09% to 66.22%) | NR | 5 (NR) | 92.55% | 2020 |
| Africa | Prevalence (critically low) (S19) | 16% (9% to 24%) | NA | 3 (NR) | NA | 2018 |
| The Americas Region | Prevalence (critically low) (S19) | 36% (23 to 50%) | NR | 3 (NR) | NR | 2018 |
| European Region | Prevalence (critically low) (S19) | 11% (4% to 21%) | NR | 3 (NR) | NR | 2018 |
| Eastern Mediterranean Region | Prevalence (critically low) (S19) | 73% (64% to 81%) | NR | 2 (NR) | NR | 2018 |
| Western Pacific Region | Prevalence (critically low) (S19) | 5% (1% to 12%) | NA | 1 (NR) | NA | 2018 |
| Southeast Asia | Prevalence (critically low) (S19) | 22% (0% to 60%) | NA | 3 (NR) | NA | 2018 |
| **Multidrug-resistant TB + anxiety** | | | | | | |
| LMICs | Prevalence (critically low) (S19) | 24% (2% to 57%) | 12%–56% | 3 (209) | 95% | 2018 |
| The Americas Region | Prevalence (critically low) (S19) | 14% (9% to 21%) | NR | 2 (NR) | NR | 2018 |
| Southeast Asia | Prevalence (critically low) (S19) | 56% (45% to 66%) | NR | 1 (NR) | NR | 2018 |
| **Multidrug-resistant TB + psychosis** | | | | | | |
| Africa | Prevalence (critically low) (S19) | 12% (8% to 17%) | NR | 5 (NR) | NR | 2018 |
| The Americas Region | Prevalence (critically low) (S19) | 11% (7% to 17%) | NR | 2 (NR) | NR | 2018 |
| European Region | Prevalence (critically low) (S19) | 6% (0% to 17%) | NR | 2 (NR) | NR | 2018 |
| Eastern Mediterranean Region | Prevalence (critically low) (S19) | 7% (1% to 17%) | NA | 1 (NR) | NA | 2018 |
| Southeast Asia | Prevalence (critically low) (S19) | 10% (5% to 17%) | NA | 2 (NR) | NA | 2018 |

*Quality rating represents the overall confidence in the results of the review, as assessed with the AMSTAR2 tool.
†Includes depression, psychological distress, Post-traumatic stress disorder or mental disorder as a composite variable.
‡Composite measure combining some or all of: treatment failure, loss to follow-up and death.
LMICs, low-income and middle-income countries; TB, tuberculosis.

**Table 4** Risk of cancer and coronary heart disease in people with TB

| Region | Outcome (quality*) | Effect size (95% CI) | Range | Number of studies (number of participants) | I² | Year |
|--------|-------------------|---------------------|-------|--------------------------------------------|-----|------|
| Upper-middle income countries | Lung cancer (low) (S23) | RR: 1.53 (1.25 to 1.87) | NR | 9 (NR) | 94.6% | 2020 |
| | Non-Hodgkin's lymphoma (low) (S23) | RR: 1.70 (1.13 to 2.56) | NA | 1 (NR) | NA | 2020 |
| | Leukaemia (low) (S23) | RR: 1.61 (1.13 to 2.29) | NA | 1 (NR) | NA | 2020 |
| LMICs | Coronary heart disease †(S55) | RR: 1.76 (1.05 to 2.95) | NA | 4 (NR) | 97% | 2020 |

*Quality rating represents the overall confidence in the results of the review, as assessed with the AMSTAR2 tool.
†Data extracted from abstract only, as we could not obtain the full text article.
LMICs, low-income and middle-income countries; TB, tuberculosis.

worse outcomes among patients with multiple additional chronic conditions. Furthermore, Chen *et al's*[31] results highlight that some combinations of comorbidities, such as the group with cardiovascular morbidity with complications, increase the risk of negative TB treatment outcomes more than others. Considering the potential multiple-way synergies between multiple chronic conditions, a systematic review of the literature on this topic is sorely needed. This evidence gap is addressed in a complementary review by our group.[33]

We did not find any systematic reviews focusing on CCDs and NCDs in people with zoonotic TB (zTB). While this type of TB was estimated to represent 1.4% of all TB cases in 2019, this number is likely to be an underestimate, as there are poor surveillance programmes, underreporting and lack of laboratory confirmation of the causative agent.[34] It is therefore not surprising that we could not find any systematic reviews synthesising studies reporting on the prevalence of comorbidities specifically in zTB.

In addition to the gaps in the literature, our meta-review also highlights the need for systematic reviews of higher quality, as most of the identified systematic reviews were assessed as low or critically low quality according to AMSTAR2, limiting the certainty we can have in their results. The systematic reviews with high or moderate quality that we have identified reported prevalence of TB + HIV in SSA (and Ethiopia), the effect of HIV in people with DR-TB in SSA, the prevalence of HIV in people with TB lymphadenitis in Africa (and Ethiopia) and the prevalence of depression in people with TB and with MDR-TB in LMICs.

### Strengths

Our review has several strengths, such as an extensive search strategy, including databases of grey literature and protocols. Considering that PROSPERO is the main registry for systematic reviews and our efforts to contact authors of potentially relevant protocols, we are confident in the coverage of our search strategy. Another strength of our review is our focus on LMICs, making sure that data from HICs was not included, as the differences in risk factors, resources and treatment opportunities would make the results less applicable to LMICs.

### Limitations

Our review has several limitations as well. First, most of the meta-analyses had very high heterogeneity and should therefore be interpreted with caution. This was the case even in systematic reviews focused on a single country. While part of this heterogeneity could be explained by methodological differences between the included studies (eg, differences in the definitions and measurement of comorbidities), it could also reflect variation, inside a country, in how TB treatment strategies are adapted to local needs, their cultural acceptance and funding limitations. As risk factors for specific CCDs and NCDs are also heterogeneous between regions (eg, prevalence of HIV

in the community, smoking habits, access to treatment, etc), the pattern of TB comorbidities is also likely to vary both between and within countries. Second, more than half of the studies summarised in our results had low or critically low quality. Third, despite the large number of systematic reviews identified in our review, our focus on LMICs excluded many results reported in them. Finally, we found little evidence regarding the burden of TB multimorbidity, which was one of the goals of this review. This highlights gaps in the body of evidence of systematic reviews, suggesting new future lines of research.

## Conclusion

Given the fact that multimorbidity is common in LMICs[35 36] and is associated with a wide range of adverse outcomes for the individual, family and society, and poses challenges for healthcare systems, particularly in LMICs, our results are important.[37 38] TB multimorbidity appears to be common and to have additional burdensome impact, deserving urgent attention.[15 39] Research is needed to identify early at-risk populations and ultimately prevent the onset of TB multimorbidity and to develop effective treatments and clinical pathways to care for this heterogeneous and burdensome group of people.[15] The high prevalence of TB multimorbidity in LMICs is a triple challenge, as these regions already have the highest (and growing) number of people with multimorbidity generally, the highest levels of TB, and health and social care systems which are stretched/sparse and unable to deal with these complexities.[15] Thus, urgent research is needed to better address this clearly prevalent, burdensome, and important issue.

**Acknowledgements** This review is part of the TB Multimorbidity Network project (https://www.impactsouthasia.com/tbmm/), which is funded by the Medical Research Council (MRC Grant reference MC_PC_MR/T037806/1).

**Contributors** NS, BS, KS and HE conceptualised the study. AJ, ER, AE, YL, RH, HE, KS, BS and NS contributed to the design of the study. AJ, ER and NS contributed to the titles and abstracts screening. AJ, ER and SA contributed to the full-text screening and data extraction. ER and SA contributed to the quality assessment. AJ summarised the results and wrote the first draft of the report with input from BS and NS. ER, SA, AE, YL and KS contributed to the interpretation of the results and revised the manuscript. AJ is responsible for the overall content as the guarantor. All authors had full access to all the data in the study and had final responsibility for the decision to submit for publication.

**Funding** This work was supported by Medical Research Council UK, grant number MC_PC_MR/T037806/1.

**Map disclaimer** The inclusion of any map (including the depiction of any boundaries therein), or of any geographic or locational reference, does not imply the expression of any opinion whatsoever on the part of BMJ concerning the legal status of any country, territory, jurisdiction or area or of its authorities. Any such expression remains solely that of the relevant source and is not endorsed by BMJ. Maps are provided without any warranty of any kind, either express or implied.

**Competing interests** None declared.

**Patient and public involvement** Patients and/or the public were involved in the design, or conduct, or reporting or dissemination plans of this research. Refer to the Methods section for further details.

**Patient consent for publication** Not applicable.

**Ethics approval** Not applicable.

**Provenance and peer review** Not commissioned; externally peer reviewed.

**Data availability statement** All data relevant to the study are included in the article or uploaded as supplementary information.

**ORCID iDs**
Alexander Jarde http://orcid.org/0000-0002-2951-9741
Rumana Huque http://orcid.org/0000-0002-7616-9596
Helen Elsey http://orcid.org/0000-0003-4724-0581
Kamran Siddiqi http://orcid.org/0000-0003-1529-7778
Najma Siddiqi http://orcid.org/0000-0003-1794-2152

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
