## [Reviewer comments · BMJ Open]

ARTICLE DETAILS

TITLE (PROVISIONAL)	Prevalence and risks of tuberculosis multimorbidity in low- and middle-income countries: a meta-review
AUTHORS	Jarde, Alexander; Romano, Eugenia; Afaq, Saima; Elsony, Asma; Lin, Yan; Huque, Rumana; Elsey, Helen; Siddiqi, Kamran; Stubbs, B; Siddiqi, Najma

VERSION 1 – REVIEW

REVIEWER	Bruchfeld, Judith Karolinska Universitetssjukhuset, Infectious Diseases
REVIEW RETURNED	02-Mar-2022

GENERAL COMMENTS	The authors have made an impressive effort to address an important research question. However findings are far from novel with both HIV, DM and mental health as risk factors or comorbidity in TB patients. Mental health issues such as depression or psychosis may also be related to treatment of both drug sensitive Tb but in particular MDR TB per se. The evidence to draw conclusions on the occurrence of for example depression in MDR TB is mainly based on studies of critically low quality. That a number of meta-analyses have critically low quality is important information but makes the overall validity of conclusions in the study questionable. My recommendation is to focus on these difficulties rather than drawing additional conclusions regarding the stated research questions.
---

REVIEWER	Zhou, Jifang China Pharmaceutical University - Jiangning Campus, School of International Pharmaceutical Business
REVIEW RETURNED	15-Mar-2022

GENERAL COMMENTS	We thank the editor of BMJ Open for the invitation to review the manuscript entitled "Prevalence and risks of tuberculosis multimorbidity in low- and middle-income countries: a meta-review". The authors conducted a systematic review of systematic reviews (meta-review) of tuberculosis multimorbidity, which defined as co-occurrence of TB with other chronic conditions by authors. Over past half century, tuberculosis epidemic has been effectively contained in coordinated initiatives by governments as well as international organizations such as WHO and the international tuberculosis association (ITA). While current researches mainly focus on the treatment and management of drug-resistant tuberculosis, the increasing non-communicable disease burdens among aging TB patients is a major concern. This study mainly summarized the most frequent chronic conditions with tuberculosis in low and middle-income countries, and provided a direction of magnitude of additional ill health burden due to chronic conditions in these
--

	countries. This manuscript is well written and easy to follow. We believe the research is much needed for contemporary TB management strategy, especially in LMIC settings. However, we have noted a number of limitations that need to be addressed before the manuscript could be accepted. Major issues 1. There are two flaws, one is the absence of Table 2, the other is conflicting statement between review results and review objectives. The authors wanted to evaluate multimorbidity with TB, to evaluate NCDs with TB and CCDs with TB, but the results only represent NCDs with TB (mental disorders). The authors should rearrange table serial number and explain why there is lack of evaluation of CCDs with TB. 2. About 80% of tuberculosis was diagnosed as pulmonary tuberculosis, while the others occurred in different parts of body, such as extrapulmonary TB, which makes TB treatment challenging, as well as the rising trend of drug-resistant tuberculosis in recent years. Different regions/countries may adopt varying TB treatment strategy based on local needs, cultural acceptance, and budgetary restrictions. Besides, different types of tuberculosis may naturally represent different patterns of multimorbidity. Due to the lack of evidence in the literature on TB or the quality of included reviews, the authors should try to explain the heterogeneity in the discussion section. 3. Zoonotic tuberculosis is a form of tuberculosis in people caused by Mycobacterium bovis, which can't be distinguished by most commonly used tests in clinical. The African region carries the heaviest burden, followed by the South-East Asian region. The authors should discuss about zoonotic tuberculosis even if there is no adequate surveillance data, such as forward an urgent need in the future direction of research in order to make patients in LMIC life better. Whether genotype-based diagnostic techniques, like GenExpert would be helpful. Minor issues 1. The prevalence of multimorbidity is rising in LMICs and high-income countries. While the NCD burdens are more pronounced in the industrialized countries, LMICs has observed drastic rise over the past decades, probably due to changes in life style, dietary and urbanization. The authors should provide clear logic in the introduction section on potential factors that have driven the changing patterns of comorbid conditions. Additionally, authors should make clear distinction on infectious comorbidity (HIV, viral hepatitis, HPV), with NCDs (diabetes, depression etc.), with more precise health losses in DALY terms, preferably describe CCD and NCD in separate paragraphs. 2. The distribution of outcomes is reasonable and accompanied by references. But the odds ratios didn't reflect on the comparison of single chronic condition and multiple chronic condition, there may be some difference in the subgroup comparison. If there is no significant statistical difference in the comparison of single chronic condition and multiple chronic condition, the authors need to mention it. 3. There is a discrepancy in Page 4, line 23. Take depression for example, from my point of view, some patients with depression may be arising after TB treatment initiation. The authors should divide the included chronic conditions or multimorbidity with complications more explicitly. 4. Currently, larger database analyses could offer insights on comorbid conditions, especially with EHR data. The authors should
--	---

	consider cite and critically assess some more recent studies: Chen Q, Che Y, Xiao Y, Jiang F, Chen Y, Zhou J, Yang T. Impact of Multimorbidity Subgroups on the Health Care Use and Clinical Outcomes of Patients With Tuberculosis: A Population-Based Cohort Analysis. Front Public Health. 2021 Oct 8;9:756717. doi: 10.3389/fpubh.2021.756717. PMID: 34692632; PMCID: PMC8531479. Soares LN, Spagnolo LML, Tomberg JO, Zanatti CLM, Cardozo-Gonzales RI. Relationship between multimorbidity and the outcome of the treatment for pulmonary tuberculosis. Rev Gaucha Enferm. 2020 Jun 5;41:e20190373. English, Portuguese. doi: 10.1590/1983-1447.2020.20190373. PMID: 32520116.
--	---

VERSION 1 – AUTHOR RESPONSE

Reviewer: 1:

1. The authors have made an impressive effort to address an important research question. However findings are far from novel with both HIV, DM and mental health as risk factors or comorbidity in TB patients. Mental health issues such as depression or psychosis may also be related to treatment of both drug sensitive Tb but in particular MDR TB per se.

>We agree with the reviewer that HIV, DM and depression as the most prevalent TB comorbidities have been previously documented. However, our review highlights that the prevalence of these conditions is not homogeneous across LMIC. Understanding how the pattern of comorbidities changes from one region to another is also important. We acknowledged both the reviewer's point and state our counterpoint by adding the following sentence in the discussion.

While HIV, DM and depression are well-known comorbidities of TB, our review highlights that their prevalence can vary, in some cases substantially, between different countries or regions. Such regional differences should be taken into account when designing interventions, illustrating how a one-size-fits-all approach is unlikely to succeed.

2.The evidence to draw conclusions on the occurrence of for example depression in MDR TB is mainly based on studies of critically low quality. That a number of meta-analyses have critically low quality is important information but makes the overall validity of conclusions in the study questionable. My recommendation is to focus on these difficulties rather than drawing additional conclusions regarding the stated research questions.

>We have further expanded the discussion of the quality of the included systematic reviews in the Discussion, emphasising the comparisons were the quality was moderate or high:

In addition to the gaps in the literature, our meta-review also highlights the need for systematic reviews of higher quality, as most of the identified systematic reviews were assessed as low or critically low quality according to AMSTAR2, limiting the certainty we can have in their results. The systematic reviews with high or moderate quality that we have identified reported prevalence of TB+HIV in SSA (and Ethiopia), the effect of HIV in people with DR-TB in SSA, the prevalence of HIV in people with TB lymphadenitis in Africa (and Ethiopia), and the prevalence of depression in people with TB and with MDR-TB in LMIC.

Reviewer: 2:

Major issues

1. There are two flaws, one is the absence of Table 2, the other is conflicting statement between review results and review objectives. The authors wanted to evaluate multimorbidity with TB, to evaluate NCDs with TB and CCDs with TB, but the results only represent NCDs with TB (mental

disorders). The authors should rearrange table serial number and explain why there is lack of evaluation of CCDs with TB.

>We thank the reviewer for pointing out the absence of Table 2 and subsequent incorrect indexing. We have now resolved this issue. We respectfully disagree with the reviewer that we have only addressed NCDs as HIV, a communicable disease, is included in the review. We acknowledge that it was not the remit of the review to include all possible communicable and noncommunicable comorbid conditions; we only selected the most common.

2. About 80% of tuberculosis was diagnosed as pulmonary tuberculosis, while the others occurred in different parts of the body, such as extrapulmonary TB, which makes TB treatment challenging, as well as the rising trend of drug-resistant tuberculosis in recent years. Different regions/countries may adopt varying TB treatment strategies based on local needs, cultural acceptance, and budgetary restrictions. Besides, different types of tuberculosis may naturally represent different patterns of multimorbidity. Due to the lack of evidence in the literature on TB or the quality of included reviews, the authors should try to explain the heterogeneity in the discussion section.

>We have expanded our discussion regarding the heterogeneity of the results:
...most of the meta-analyses had very high heterogeneity and should therefore be interpreted with caution. This was the case even in systematic reviews focused on a single country. While part of this heterogeneity could be explained by methodological differences between the included studies (e.g. differences in the definitions and measurement of comorbidities), it could also reflect variation, inside a country, in how TB treatment strategies are adapted to local needs, their cultural acceptance and funding limitations. As risk factors for specific CCDs and NCDs are also heterogeneous between regions (e.g. prevalence of HIV in the community, smoking habits, access to treatment, etc.), the pattern of TB comorbidities is also likely to vary both between and within countries.

3. Zoonotic tuberculosis is a form of tuberculosis in people caused by *Mycobacterium bovis*, which can't be distinguished by most commonly used tests in clinical. The African region carries the heaviest burden, followed by the South-East Asian region. The authors should discuss about zoonotic tuberculosis even if there is no adequate surveillance data, such as forward an urgent need in the future direction of research in order to make patients in LMIC life better. Whether genotype-based diagnostic techniques, like GenExpert would be helpful.

>We thank the reviewer for bringing zTB to our attention. We have highlighted the lack of any systematic reviews focusing on comorbidities in people with zTB:
We did not find any systematic reviews focusing on CCDs and NCDs in people with zoonotic TB (zTB). While this type of TB was estimated to represent 1.4% of all TB cases in 2019, this number is likely to be an underestimate, as there are poor surveillance programmes, under-reporting and lack of laboratory confirmation of the causative agent (REF). It is therefore not surprising that we could not find any systematic reviews synthesising studies reporting on the prevalence of comorbidities specifically in zTB.

Minor issues

1. The prevalence of multimorbidity is rising in LMICs and high-income countries. While the NCD burdens are more pronounced in the industrialized countries, LMICs has observed drastic rise over the past decades, probably due to changes in life style, dietary and urbanization. The authors should provide clear logic in the introduction section on potential factors that have driven the changing patterns of comorbid conditions.

>We have expanded the introduction section as suggested by the reviewer:

Multimorbidity is a growing global concern and its prevalence is rising in low- and middle-income countries (LMICs), as CCDs such as tuberculosis (TB) and HIV remain major public health issues. and where NCDs are increasing due to major demographic shifts, urbanization, changing environmental factors, economic empowerment and accompanying lifestyle changes and where CCDs such as tuberculosis (TB) and HIV remain major public health issues. This shift away from risks for CCD in children towards those for NCD in adults is also reflected in the steady increase in the burden of disability-adjusted life years (DALYs) attributed to NCDs over the past decades, reaching 34% in low-income countries, and up to 82% in middle-high-income countries in 2019.

1b. Additionally, authors should make clear distinction on infectious comorbidity (HIV, viral hepatitis, HPV), with NCDs (diabetes, depression etc.), with more precise health losses in DALY terms, preferably describe CCD and NCD in separate paragraphs.

>We have added information on DALYs for the main TB comorbidities.

TB frequently co-occurs with NCDs, including diabetes mellitus (DM, 2.79% of worldwide DALYs), depression (1.84% of worldwide DALYs), and cancer (neoplasms representing 9.93% of worldwide DALYs). Depression and DM have been reported to be important risk factors for TB. Similarly, CCDs such as HIV (1.88% of worldwide DALYs) and TB adversely affect each other at the molecular, cellular, individual and population levels.

2. The distribution of outcomes is reasonable and accompanied by references. But the odds ratios didn't reflect on the comparison of single chronic condition and multiple chronic condition, there may be some difference in the subgroup comparison.

If there is no significant statistical difference in the comparison of single chronic condition and multiple chronic condition, the authors need to mention it.

>We agree with the reviewer regarding the importance to explore the differences between having one or multiple additional chronic conditions in people with TB. However, none of the systematic reviews reported results on the prevalence and/or associated risks of more than one additional chronic condition and this issue could not be explored.

In the RESULTS section, we noted that:

None of the systematic reviews reported results on the prevalence and/or associated risks of more than one additional chronic condition in people with TB.

In the Discussion section:

No systematic review pooled the prevalence of two or more additional chronic conditions in people with TB and differences between people with TB and a single chronic condition vs multiple additional chronic conditions could not be explored.

3. There is a discrepancy in Page 4, line 23. Take depression for example, from my point of view, some patients with depression may be arising after TB treatment initiation.

The authors should divide the included chronic conditions or multimorbidity with complications more explicitly.

>The reviewer makes a good point and we can see how the original wording could be misleading. We have replaced the sentence they allude to with the following:

Conditions considered side effects of TB medications, such as nausea or diarrhoea, were not considered chronic conditions for this review.

As suggested by the reviewer, we have clearly separated the list of conditions included and not included in Supplementary Table 5.

4. Currently, larger database analyses could offer insights on comorbid conditions, especially with EHR data. The authors should consider cite and critically assess some more recent studies:

Chen Q, Che Y, Xiao Y, Jiang F, Chen Y, Zhou J, Yang T. Impact of Multimorbidity Subgroups on the Health Care Use and Clinical Outcomes of Patients With Tuberculosis: A Population-Based Cohort Analysis. *Front Public Health*. 2021 Oct 8;9:756717. doi: 10.3389/fpubh.2021.756717. PMID: 34692632; PMCID: PMC8531479.

Soares LN, Spagnolo LML, Tomberg JO, Zanatti CLM, Cardozo-Gonzales RI. Relationship between multimorbidity and the outcome of the treatment for pulmonary tuberculosis. *Rev Gaucha Enferm*. 2020 Jun 5;41:e20190373. English, Portuguese. doi: 10.1590/1983-1447.2020.20190373. PMID: 32520116.

>We thank the reviewer for bringing these studies to our attention. We have referenced them in our Discussion:

...the lack of systematic reviews focused on people with TB and more than one additional chronic condition, which is an increasingly likely scenario as the prevalence of NCDs in LMICs grows.[2] In this regard, several cohort studies have assessed the impact of multimorbidity on TB treatment outcomes, such as in China or Brazil, finding worse outcomes among patients with multiple additional chronic conditions. Furthermore, Chen et al (2021) results' highlight that some combinations of comorbidities, such as the group with cardiovascular morbidity with complications, increase the risk of negative TB treatment outcomes more than others.

VERSION 2 – REVIEW

REVIEWER	Zhou, Jifang China Pharmaceutical University - Jiangning Campus, School of International Pharmaceutical Business
REVIEW RETURNED	11-Aug-2022
GENERAL COMMENTS	I feel that the authors have satisfactorily addressed the issues raised by the reviewers and the manuscript looks good for publication purposes. I congratulate the authors on this comprehensive work in tuberculosis.